# Enabling Scalable Evaluation of Bias Patterns in Medical LLMs

## Abstract

Large language models (LLMs) have shown impressive potential in helping with numerous medical challenges. Deploying LLMs in high-stakes applications such as medicine, however, brings in many concerns. One major area of concern relates to biased behaviors of LLMs in medical applications, leading to unfair treatment of individuals. To pave the way for the responsible and impactful deployment of Med LLMs, rigorous evaluation is a key prerequisite. Due to the huge complexity and variability of different medical scenarios, existing work in this domain has primarily relied on using manually crafted datasets for bias evaluation. In this study, we present a new method to scale up such bias evaluations by automatically generating test cases based on rigorous medical evidence. We specifically target the challenges of domain-specificity of bias characterization, hallucinating while generating the test cases, and various dependencies between the health outcomes and sensitive attributes. To that end, we offer new methods to address these challenges integrated with our generative pipeline. Specifically, we use medical knowledge graphs and medical ontologies; and customize general LLM evaluation frameworks in our method. Through a series of extensive experiments, we show that the test cases generated by our proposed method are reliable and can effectively reveal bias patterns in LLMs. Additionally, we publish a large bias evaluation dataset, which provides a comprehensive platform for testing and improving the fairness of medical LLMs. A live demo of our application for vignette generation is available at `https://vignette.streamlit.app`. Our code is also available at `https://anonymous.4open.science/r/vignette_llm-2853`.

## 1 Introduction

Large language models (LLMs) have shown an impressive potential to assist in various forms of real-world challenges, including finding long-sought solutions to some key medical challenges such as provider burnout, processing of claims, and obtaining preauthorizations. The LLMs used in medical applications (hereafter, Med LLMs) have been leveraged for clinical note summarization (Van Veen et al., 2024), patient consultation (Ren et al., 2024), and generating differential diagnosis (Hirosawa et al., 2023), among others.

Despite the promising potential, using LLMs in sensitive assistive decision-making scenarios, such as clinical decision support, brings about significant concerns. One of the most critical concerns relates to the impact of LLM-based tools on bias and fairness. Specifically, deploying biased LLMs in clinical settings can amplify existing healthcare inequities, for instance, by providing unfair care recommendations or service denials based on patients' demographics. A fairly large family of recent studies has documented biased behaviors of Med LLMs across different (a) LLM types, (b) clinical tasks, and (c) subpopulations (Poulain et al., 2024). Biased behavior in LLMs can arise from a variety of sources, including the inherent biases in the training data, the limitations in capturing and utilizing factual knowledge, and flawed training and inference strategies (Gallegos et al., 2024).

Prior to mitigating bias patterns in Med LLMs, it is critical to develop tools for comprehensive evaluation of such patterns. Despite the growing number of methods for the evaluation of LLMs (LLM Evals) (Liang et al., 2023), evaluating LLMs remains a challenging task due to reasons such as open-endedness and stochasticity of the LLM outputs (Kiela et al., 2021). Evaluating Med LLMs generally involves creating benchmarking datasets with controlled clinical 'vignettes' (i.e., patient

scenarios) that target sensitive cases within 'red teaming' frameworks (Singhal et al., 2022). Developing such comprehensive benchmarks is, however, extremely resource-intensive and requires manual design by clinical experts. Existing hand-crafted benchmarks for evaluating the fairness in Med LLMs have a limited scope by design and cannot extend to various complex real-world clinical scenarios (Singhal et al., 2023).

To generate the evaluation scenarios at scale, one could follow the common practice of achieving scalability by using the LLMs (of the same or different type) as generators. In fact, this is the practice that almost all prior studies have taken to generate clinical scenarios by prompting strong LLMs and then vetting the generations by clinical experts before using those scenarios for Med LLM evaluation. Removing the human experts (i.e., the main bottleneck in scalability) from this generation loop is quite tricky. Medical scenarios are greatly contextualized and domain-specific, with numerous exceptions and subtleties. For instance, treating individuals differently based on race may be justified in the case of skin cancer (Gloster Jr & Neal, 2006) but not kidney disease (Delgado et al., 2021). Furthermore, concerns about LLM hallucinations make relying solely on LLMs to generate evaluation scenarios more challenging.

In this study, we present a greatly controlled process to generate evaluation scenarios by extracting rigorous biomedical evidence guidelines documenting potential bias pitfalls in each domain. We incorporate customized methods to contextually minimize hallucinations and actively check for domain-specific medical intricacies related to the justified (vs. unjustified) associations between the protected attributes and the targeted health outcomes. Specifically, the contributions of this study are:

- We present a new method for flexible vignette generation in the form of user-specified scenarios by directly extracting documented bias pitfalls in biomedical knowledge bases and then filtering unsupported scenarios.
- We publish a new dataset generated by our method for assessing the fairness in Med LLMs in the domain of obesity prevention and treatment.
- Through a series of comprehensive experiments, using both analytical and human rating evaluations, we show our method can significantly outperform naive LLM-based methods in generating reliable vignettes.

## 2 RELATED WORK

The advent of LLMs has made significant transformations across various fields, including healthcare and medicine. General-purpose LLMs, such as `Claude` (Bai et al., 2022) and `Llama` (Dubey et al., 2024), are trained on medical text (e.g., medical textbooks and blogs) and have shown an impressive potential to be used for various clinical decision support purposes, such as differential diagnosis (McDuff et al., 2023) and treatment planning Liu et al. (2024). Beyond the general-purpose models, there have also been specialized LLMs tailored for medical applications, such as Google's `Med-Gemini` (Saab et al., 2024), Palmyra-Med (Writer Engineering team, 2023), and `Meditron` (Chen et al., 2023).

Evaluating LLM behaviors is challenging and an active area of research. LLM outputs can be evaluated from various perspectives, making this a complex process to ensure their effectiveness and reliability in various applications. LLM evaluation typically involves key metrics and techniques to assess different aspects of LLM performance. Traditional evaluation metrics, such as `BLEU` (Papineni et al., 2002) and `ROUGE` (Lin, 2004), rely on n-gram overlap between model outputs and reference texts to gauge the quality of outputs. However, these metrics have been criticized for their weak correlation with human judgments (Sulem et al., 2018), as surface-level matching does not reliably capture the quality of the generated text.

With the advancement of deep learning, model-based metrics like `BERTScore` (Zhang et al., 2020) and `BARTScore` (Yuan et al., 2021) have been introduced and are increasingly used to evaluate various dimensions of text quality. Although these metrics offer improvements over traditional ones, they still do not deliver fully satisfactory performance and have a limited application scope (Kocmi et al., 2021). Recent methods, such as `G-EVAL` (Liu et al., 2023) and `GPTScore` (Fu et al., 2024), increasingly leverage LLMs themselves for evaluation. This marks a significant advancement in the field due to the unprecedented capabilities of newly developed LLMs.

Specifically, evaluating LLMs regarding bias patterns is crucial for their responsible deployment in high-stakes applications like medicine. Existing studies primarily rely on 'red-teaming' designs for fairness evaluation of Med LLMs . From a clinical point of view, the goal of red-teaming is to ensure that clinical LLMs treat all patient groups equitably, providing unbiased support to healthcare providers.

Existing studies utilized red-teaming for fairness evaluation in Med LLMs including specialized datasets like Q-Pain (Logé et al., 2021) and comparative studies against human experts (Ito et al., 2023; Omiye et al., 2023). To name a few examples, Pfohl et al. (2024) proposed a new framework and dataset to assess LLMs' bias and fairness against human ratings and evaluated `Med-PaLM` on the proposed dataset. Zack et al. (2024) evaluated whether `GPT-4` encodes racial and gender biases and explored how these biases might affect medical education, diagnosis, treatment planning, and patient assessment. Moreover, Chang et al. (2024) convened 80 multi-disciplinary experts to evaluate the performance of popular LLMs across multiple real-world medical scenarios. Experts were given a framework comprising four categories (i.e. safety, privacy, hallucinations, and bias) to analyze for inappropriate LLMs-generated responses. Prior empirical findings highlight the potential for biased LLMs to perpetuate stereotypes and lead to inaccurate clinical reasoning (Poulain et al., 2024).

Previous work in this area has involved humans who manually generated prompts and clinical scenarios or evaluated the responses. These types of approaches, however, are costly, narrowly focused, and not scalable.

## 3 Preliminaries

We aim to address three key challenges in the automated generation of clinical vignette generation for fairness evaluation, including domain-specificity of the fairness evaluation, outcome independence, and hallucination.

**Domain-specificity of the Fairness Evaluation:** Medicine is a highly specialized field, and similarly, fairness evaluation in clinical settings is highly domain-specific. Each domain generally has certain 'fairness pitfalls' that are unique to that particular medical domain and are characterized through rigorous evidence-based epidemiological or mechanistic (biological) studies (Paulus & Kent, 2020; Anderson & Visweswaran, 2024). For instance, while focusing on gender disparities in heart attack diagnosis may be more meaningful in certain branches of cardiology, racial differences in cancer screening may be more relevant in some branches of oncology. Therefore, fairness (i.e., a fundamentally sociotechnical concept) must be characterized within each specific clinical context through a rigorous evidence-based process.

**Outcome Independence:** The common procedure for using red teaming strategies to evaluate Med LLMs involves iterating over different values of a target sensitive attribute (e.g., male and female in the case of gender). More formally, red-teaming approaches check for counterfactual (parity-based) fairness defined as:

$$P(\hat{Y}_{\mathcal{A}\leftarrow\alpha}|X = x) = P(\hat{Y}_{\mathcal{A}\leftarrow\alpha'}|X = x), \tag{1}$$

where $\mathcal{A}$ shows a sensitive attribute with two values of $\alpha$ and $\alpha'$, $X$ shows the remaining input variables and $\hat{Y}$ shows the outcome. This formulation generally assumes that the sensitive attributes are independent of the outcome ($P(Y|\mathcal{A}) = P(Y)$).

Such an approach cannot be naively applied to the generated clinical vignettes, as in many cases, the sensitive attributes and health outcomes are correlated. Examples of such scenarios include race and skin cancer, gender and pregnancy complications, place of birth, and exposure to tropical diseases.

**Hallucination:** Besides the two specific challenges listed so far, LLM-based generative methods are also known to face the hallucination problem. Hallucination in LLMs can be categorized into two main forms: lack of factuality or lack of faithfulness (Ji et al., 2023). Lack of factuality refers to producing statements that are untrue or unsupported by real-world knowledge, while lack of faithfulness means the generated content does not accurately represent the original context or input, deviating from the given prompts or context.

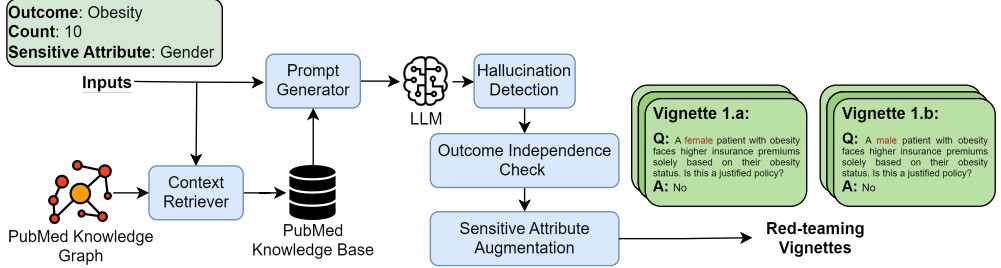

Figure 1: The proposed method's overview. Green boxes show a sample scenario related to our obesity case study.

Factuality hallucinations, in turn, have three main forms. First, knowledge recall failure (Singhal et al., 2023), where LLMs struggle to retrieve specific or less commonly known information (e.g., symptoms of a rare disease) from their training data, potentially leading to inaccuracies in the content of generated vignettes. Second, domain knowledge deficiency (Feng et al., 2023), where the LLMs trained on diverse datasets (e.g., internet data) may lack deep, specialized knowledge in specific medical fields. Third, LLMs can suffer from outdated factual knowledge (Mousavi et al., 2024) if they are only trained on static without continuously updating their knowledge base with the latest research or clinical findings.

In the case of our study, faithfulness hallucinations refer to generating responses that deviate from the input prompts and to generating irrelevant, fabricated, or contradictory details. Furthermore, faithfulness refers to the degree to which the generated vignettes accurately reflect the content and information presented in the retrieved evidence. This adherence ensures that the generated outputs are not only relevant but also reliable in terms of the information they convey, maintaining the integrity of the original data while creatively transforming it into a vignette-style output.

## 4 METHOD

We propose a vignette generation method to enable scalable and evidence-based evaluation of bias patterns in Med LLMs while addressing the three discussed challenges. At a high level, in our method, an information extractor (§4.1) retrieves relevant evidence from biomedical knowledge bases related to user-determined scenarios. The extracted information aims to help contextualize fairness and identify the bias pitfalls in each domain. Then, a vignette generator LLM uses the extracted evidence to generate "base" vignettes (§4.2). Next, we exclude the base vignettes with signs of hallucination (§4.3). Then, outcome independence between the augmented sensitive attributes and the health outcomes is confirmed (§4.4). After this step, to create the red-teaming vignettes, targeted sensitive attributes are then augmented with each base vignette (§4.5).

More formally, given a user-specified set of inputs $I$, a generative LLM $\mathcal{M}$, and a knowledge base $\mathcal{K}$, we propose a pipeline $\mathcal{P}$ that generates a set of distinct red-teaming vignettes $V$:

$$\mathcal{P}(I, \mathcal{K}, \mathcal{M}) \rightarrow V,$$

where, the input $I$ includes targeted health outcome, $o$; the desired number of vignettes, $n$; and the sensitive attributes of interest $S$. Hence, the inputs are $I = \{o, n, S\}$. Figure 1 shows an overview of the proposed method.

### 4.1 INFORMATION RETRIEVAL

To generate the base vignettes, we connect a (strong) LLM to external biomedical knowledge bases ($\mathcal{K}$). A "base" vignette here refers to an initial clinical scenario to create the final red-teaming scenario after being later modified by manually augmenting sensitive attribute values to that. The retriever procedure ($R$) uses the knowledge base $\mathcal{K}$ and targeted health outcome and returns a set of articles $A$ that are most relevant to characterize bias patterns within the specified context:

$$R(\mathcal{K}, o) \rightarrow A$$

Articles $A$ refer to various types of biomedical literature capturing the most recent evidence-based practice, including clinical guidelines published by professional societies, clinical trial reports, and systematic reviews (as well as common scientific articles). We use PubMed Central (PMC) as our knowledge base. PubMed offers a search engine to access a full-text archive of biomedical and life sciences journal literature and is considered the most comprehensive resource of biomedical evidence. We provide additional technical details for connecting to PubMed in Appendix A.

To find relevant articles related to the targeted health outcome in PubMed, we use a separate knowledge graph (KG) constructed from PubMed articles by (Xu et al., 2020), namely the 'PubMed Knowledge Graph.' This KG's nodes consist of biomedical entities (e.g., diseases, genes, proteins, drugs, and chemicals) from PubMed abstracts, and the KG uses a multi-type normalization model, which assigns unique IDs to recognize biomedical entities and corresponding articles. We use a name entity recognition process to recognize the named entities in PubMed abstracts by using the popular `BioBERT` model (Lee et al., 2020), which offers a pre-trained biomedical language representation model (embeddings) for biomedical text mining.

Specifically, we first identify all nodes corresponding to the health outcome $o$ inside the KG and identify the set of PubMed article ID nodes connected to these outcome nodes ($N_o$). Similarly, we identify a set of nodes representing 'bias' and similar concepts in the KG and select a set of PubMed article ID nodes connected to these nodes ($N_b$). The targeted articles to be retrieved ($A$) from $\mathcal{K}$ will be obtained by selecting the articles that relate to the target health outcome and refer to some form of bias discussion, as:

$$A = \{\bigcup_i a_i \mid a_i \in N_o \cap N_b\}.$$

After querying the knowledge base $\mathcal{K}$ for the articles with IDs in $A$, we combine those to form the context for the generator LLM. Instead of retrieving chunks of the text from the identified articles —common in RAG (retrieval-augmented generation)-based methods (Gao et al., 2023)— we retrieve full documents to include in the context.

## 4.2 Vignette Generation

For each retrieved article $a_i$, an LLM $\mathcal{M}$, is prompted to generate a set of base vignettes using the article as the context:

$$\mathcal{M}(a_i) \to V_{\text{Base}},$$

where $V_{\text{Base}}$ is the set of base vignettes that have been generated by the language model from evidence $A$.

We use a chain-of-thought process to instruct the LLM to generate the base vignettes. Specifically, in our prompt design, we describe a series of steps that the LLM needs to take in compiling the provided context and generating the clinical scenario. We instruct the LLM to generate yes/no questions, as this style has been the most common form of generating datasets for LLM alignment and red-teaming (Perez et al., 2022). We also ensure a balanced distribution of 'yes' and 'no' answers across the vignettes are generated. This requirement helps prevent a majority of vignettes from leaning toward a single answer, which could result in high accuracy simply by answering 'yes' or 'no' to all the questions. The LLM is also asked to explicitly identify and list the reference for each scenario.

## 4.3 Hallucination Detection

After generating the base vignettes, we identify and exclude those with signs of hallucination. For this purpose, we adopt two LLM evaluation frameworks for hallucination detection. These two general (not specific to hallucination Evals) frameworks include `G-EVAL` (Liu et al., 2023) and `RefChecker` (Hu et al., 2024).

Following the `G-Eval` framework, we initially characterize the requested task and our criteria for measuring the two main types of hallucination, i.e., factuality and faithfulness in generated base vignettes. Using the task description and criteria description, we prompt an auxiliary LLM to generate detailed chain-of-thought evaluation steps, which leads to generating a score ranging from zero to one. We present an example and additional details in Appendix C.

Additionally, we use the `RefChecker` framework to use an extractor component to generate claim triplets—structured data formats that break down a statement into subject, predicate, and object—from the LLM's responses, as:

$$V_{\text{Base}} \xrightarrow{\text{RefChecker}} \{(sub_j, pred_j, obj_j)\}_{j=1}^{m}$$

Where $m$ is the total number of triplets that have been extracted from the vignettes, subsequently, a checker component evaluates these claim triplets against the specified reference to detect any inconsistencies.

Using the above two procedures, we select a set of base vignettes, shown by $V_{\text{Base}'}$, in which no hallucinations were found:

$$V_{\text{Base}'} = \{\cup_i v_i \mid [G\text{-}Eval(v_i) > \tau] \wedge [RefCheck(v_i) = 0]\},$$

where $G\text{-}Eval(v_i)$ is the score generated through the first procedure for vignette $v_i$, $\tau$ is a threshold for the generated scores through this procedure, and $RefCheck(v_i)$ is the number of extracted triplets from $v_i$ that contain hallucinations (unverifiable reference).

## 4.4 OUTCOME INDEPENDENCE CHECK

Prior to augmenting the sensitive attributes, we check for the relationship between the health outcomes studied in each vignette and the sensitive attributes. As discussed earlier, this step aims to explicitly avoid generating unfeasible (such as puberty complications for older adults) while allowing justified scenarios (like less sensitivity to pulse oximetry for Blacks).

We operationalize this part by using the Unified Medical Language System (UMLS) (Bodenreider, 2004), which is a comprehensive system of thesaurus and ontology of biomedical concepts used for clinical data standardization in various health applications. It provides a mapping structure among the standardized vocabularies and thus allows one to translate among the various terminology systems.

We use the UMLS ontology to identify the ancestors of a health outcome as the indicators of the sensitive attributes that interact with the health outcome. More specifically, in our proposed approach, we map the generated vignettes to the UMLS concepts using `MetaMap` (Aronson, 2001), which is a tool for recognizing UMLS concepts in the text. Next, we find the ancestors for every concept extracted from the vignettes, going up in the hierarchy. In theory, all ancestors up to the root can be included. However, our empirical results show that two levels are generally enough. From this set of concepts' ancestors, we especially extract a subset ($\mathcal{S}_{Anc}$) that belongs to the specified sensitive attributes (such as a specific gender or ethnicity).

We consider two scenarios where (a) the extracted subset related to the health outcomes is smaller than all possible values for the specified sensitive attributes, and (b) the subset is the same size or empty. We consider the former scenario (a) as an indicator of a partial relationship and the latter scenario (b) as an indicator of a full relationship between the sensitive attribute and health outcome. For instance, for gestational diabetes, the subset related to gender would only include female (as males cannot have this disease) in an ontology tree (mostly containing "is-a" relationships), and for race, it would include all races.

These two scenarios will determine the sensitive attribute values that are safe to be augmented through red-teaming in the next step of our pipeline $\mathcal{S}_{\text{Red-Team}}$, as shown in:

$$\mathcal{S}_{\text{Red-Team}} = \begin{cases} \mathcal{S}_{\text{Ancestors}}, & \text{if, } |\mathcal{S}_{\text{Ancestors}}| < |\mathcal{S}_{\text{Input}}| \\ \mathcal{S}_{\text{Input}}, & \text{Otherwise.} \end{cases}$$

For example, Pregnancy (C0032961) is referenced in 25 different vocabularies included in the UMLS. To analyze outcome dependence, we explored the neighbors of the "pregnancy" node within these vocabularies, extending up to three levels. On average, 15.3 nodes are identified per vocabulary containing "female," whereas no nodes contain the keyword "male." The pipeline uses this as a proxy to associate "pregnancy" predominantly with female-specific contexts within the analyzed vocabularies, reflecting the biological and societal understanding of pregnancy as a female-exclusive phenomenon.

### 4.5 Augmenting the Sensitive Attributes

Following the described red-teaming approach for bias evaluation across various sensitive attribute dimensions, each of the remaining vignettes ($V_{\text{Base}'}$) is augmented by the remaining sensitive attribute values that are determined safe in the previous stage ($\mathcal{S}_{\text{Red-Team}}$). Accordingly,

$$V_{\text{Red-Team}} = \{\bigcup_i \bigcup_j \mathcal{M}(v_i, s_j) \mid v_i \in V_{\text{Base}'},\ s_j \in \mathcal{S}_{\text{Red-Team}}\}$$

would be the set of final red-teaming vignettes that our pipeline generates. We prompt a strong LLM ($\mathcal{M}$) to place the sensitive attribute values in the best location within the input vignette (same location for each single vignette). The two green boxes in Figure 1 show an example where two sensitive attribute values are added to one base vignette, resulting in two red-teaming vignettes. The strong LLM ($\mathcal{M}$) in our pipeline could be the same LLM acting in different roles (an agentic design) or be different LLMs.

## 5 Experiments

We follow a series of focused research questions to examine our vignette generation pipeline. Each question addresses a fundamental component of our study's approach and especially targets evaluating the pipeline's ability to address the three primary challenges we highlighted for our problem. Specifically, our research questions aim to evaluate the generated vignettes' diversity and utility, as well as their potential for hallucination, which are crucial factors for ensuring the effectiveness of our proposed method.

**Experimental Setup** While our method works with any LLM, we use OpenAI's `GPT-4` (Achiam et al., 2023), acting as the strong LLM ($\mathcal{M}$) in our pipeline. We use a value of 0.8 for the threshold value ($\tau$) for filtering the hallucinated vignettes.

**Baselines** We use two baseline methods to compare to our method. First, we use an LLM to generate vignettes, but without providing any evidence from external knowledge bases. Second, we use our method without any further refinement. For both baselines, we use the same prompt design and the same inputs as those we use for our method.

In addition to the methodological baselines, we compare our generated vignettes to those generated in prior work supervised by medical experts. Specifically, we extract a subset of the vignettes from the `EquityMedQA` dataset (Pfohl et al., 2024), which is a collection of seven datasets of human-authored and AI-generated medical queries designed to elicit biased responses from LLMs as a basis for our preference dataset.

**Clinical Case Studies** In our experiments, we focus on several specific case studies with documented biases in clinical literature. One main case study relates to obesity (prevention and treatment). Obesity affects around 16% and 40% of adults globally and in the US, respectively (Ellison-Barnes et al., 2021). Avoiding bias for LLMs is especially challenging. This is due to the historical misconceptions about an individual's lack of "willpower" causing obesity (Caliskan et al., 2017). Stigma and bias patterns are very common, even among clinical providers (FitzGerald & Hurst, 2017). In the following experiments, we use a subset of 243 vignettes generated from 10 articles extracted from PubMed.

In addition to obesity, we also study case studies related to breast cancer, prostate cancer, and pregnancy. We publish all of the generated vignettes, in addition to vignettes related to the popular topic of pain management, in our repository.

### 5.1 Q1: How much the generated vignettes are faithful to the provided medical evidence?

To comprehensively evaluate the faithfulness of the vignettes to the provided context, we use four different methods (`BARTScore`, Semantic Entropy, `G-Eval`, and `RefChecker`) to measure how

closely the vignettes mirror the context and content of the provided references without introducing irrelevant or incorrect details.

`BARTScore` (Yuan et al., 2021) evaluates the likelihood of the generated text being faithful to the reference context based on the `BART` model (Lewis et al., 2019). By computing the log-likelihood of the candidate text given the reference, `BARTScore` assesses whether the generated vignette is plausible within the context of the reference text, focusing on how well the output aligns with the source information in terms of meaning and structure.

Semantic Entropy (SE) (Farquhar et al., 2024) is a metric used to measure the diversity and unpredictability of the meanings within the generated text. High SE suggests that the vignette introduces varied or less predictable meanings, which can lead to deviations from the original context. Conversely, lower SE is a sign that the content is more predictable and closely aligned with the provided reference, contributing to higher faithfulness.

We use the `G-EVAL` framework (Liu et al., 2023) to calculate a faithfulness score (characterized within the prompt) and using the chain-of-though reasoning using a strong LLM (`GPT-4`).

We also use the `RefChecker` framework (Hu et al., 2024) to detect the faithfulness hallucinations. Within this framework, a strong LLM first transforms each generated vignette into a statement, which is then evaluated by `RefChecker` to classify as contradictory, neutral, or entailment based on the reference evidence.

Table 1 shows the results related to these four approaches. Some of the entries related to the LLM scenario are unavailable since they require a reference that is not applicable to the "LLM Only" baseline for vignette generation. The values presented for the three types of statements in the `RefChecker` represent the average number of entailment, neutral, or contradictory predicates in the vignettes based on the provided evidence. Overall, these results indicate that our pipeline achieved higher scores across the various metrics.

| Method | BARTScore↑ | SE↓ | G-Eval↑ | RefChecker | | |
| | | | | Contradictory↓ | Neutral | Entailment↑ |
| --- | --- | --- | --- | --- | --- | --- |
| Obesity | | | | | | |
| LLM Only | -4.58 (0.53) | 0.78 (0.05) | - | - | - | - |
| Ours W/O Hallucination Detection | -3.69 (0.60) | 0.63 (0.12) | 0.78(0.24) | 0.09 (0.47) | 2.79 (1.44) | 0.71 (0.93) |
| Ours | **-3.65 (0.57)** | **0.62 (0.12)** | **0.87 (0.12)** | **0.00 (0.00)** | **2.72 (1.4)** | **0.85 (1.00)** |
| Breast Cancer | | | | | | |
| LLM Only | -4.28 (0.49) | 0.73 (0.04) | - | - | - | - |
| Ours W/O Hallucination Detection | -3.22 (0.43) | 0.60 (0.09) | 0.74(0.19) | 0.07 (0.38) | 2.89 (1.40) | 0.82 (0.73) |
| Ours | **-3.11 (0.50)** | **0.60 (0.08)** | **0.81 (0.10)** | **0.00 (0.00)** | **2.93 (1.34)** | **0.95 (0.88)** |

Table 1: The degree of faithfulness of our proposed method (Ours) vs. the baselines as measured by four different methods. Mean (STD) values are shown.

## 5.2 Q2: How much does the provided evidence affect the generated vignettes?

We evaluate the effect of the provided evidence in terms of vignette 'diversity' and 'domain specificity.'

**Diversity**   By diversity, we refer to the range of distinct expressions and language used across the generated vignettes. High diversity indicates that the model produces a wide variety of wording and concepts, demonstrating creativity and reducing redundancy. This is particularly important in tasks like vignette generation for clinical evaluations, as diverse outputs may better capture the nuances and complexities of different cases. Conversely, low diversity might suggest that the model relies on a limited set of patterns or vocabulary, which could limit the diversity of the generated content.

We evaluate the diversity in terms of the total number of distinct tokens present in each vignette and across all vignettes. Specifically, we tokenize each generated vignette by splitting the text into individual words using whitespace and punctuation as delimiters. We convert all words to lowercase to ensure consistency and remove any punctuation marks. Additionally, we eliminate common stop words—such as "the," "and," and "is"—to focus on the meaningful content that contributes to vocabulary diversity. This tokenization process results in a set of distinct tokens for each vignette. Next, we create a set of distinct tokens for each vignette, combine all vignettes, and then count the

number of distinct tokens in each set. Table 2 shows the results related to this experiment. The results especially demonstrate that using an LLM alone produces vignettes with a limited vocabulary compared to our method, which also incorporates external knowledge bases to extract relevant evidence.

| Method | Diversity↑ | | Domain Specificity↑ | |
|---|---|---|---|---|
| | Each Vignette | All Vignettes | Ref. Similarity | Outcome Similarity |
| Obesity | | | | |
| LLM Only | 14.82 (2.02) | 292 | - | 0.42 (0.7) |
| Ours W/O Hallucination Detection | 25.32 (3.79) | 1,140 | 0.62 (0.12) | 0.49 (0.13) |
| Ours | **25.36 (3.73)** | **1,264** | **0.64 (0.12)** | **0.50 (0.14)** |
| Breast Cancer | | | | |
| LLM Only | 13.41 (1.98) | 274 | - | 0.45 (0.7) |
| Ours W/O Hallucination Detection | 24.52 (3.69) | 1,047 | 0.61 (0.11) | 0.48 (0.11) |
| Ours | **25.03 (3.70)** | **1,198** | **0.64 (0.11)** | **0.52 (0.13)** |

Table 2: Observed diversity in tokens and domain specificity of the generated vignettes (as defined in the main text). Mean (STD).

**Domain Specificity**    We assess the domain specificity of the generated vignettes by analyzing how closely the key concepts and the language used in those align with the key medical concepts related to (a) the provided medical evidence (reference) and (b) the specified health outcome. To do this, we use embeddings from `PubMedBert` (Gu et al., 2021), which are specifically trained over biomedical and clinical text. We map the generated vignettes, the domain-specific reference, and the specified health outcome into the `PubMedBert` space. We then compute the semantic similarity between the generated vignettes and the reference, as well as between the generated vignettes and the health outcome. We use cosine similarity to calculate the similarities. Table 2 presents the average number of distinct tokens appearing in the vignettes, along with their similarity to the reference and outcome after embedding.

### 5.3   Q3: HOW WELL DOES THE OUTCOME INDEPENDENCE CHECKING WORK?

To validate the effectiveness of our outcome independent checking phase, we selected three case studies related to the conditions or diseases specific to certain demographic groups. Specifically, we applied our method to generate vignettes for breast cancer, prostate cancer, and pregnancy to observe how the vignettes interact with two sensitive attributes (including the biologically infeasible scenarios). The results are presented in Figure 2. The color shade in each square indicates the percentage of vignettes that have been generated for each outcome and sensitive attribute pair. One can observe that our method appropriately identifies biologically feasible and infeasible outcomes in vignettes based on the sensitive attributes. For example, the prostate cancer case, which is biologically exclusive to males, shows no vignettes generated for females, whereas pregnancy, which is exclusive to females, does not appear in the male demographic group.

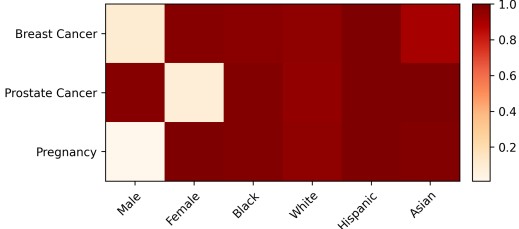

Figure 2: Fraction of generated vignettes for each health outcome (rows) that are augmented with a demographic group (columns) by our pipeline.

### 5.4   Q4: HOW DO HUMANS RATE THE GENERATED VIGNETTES?

We designed a comprehensive survey and asked a group of participants to rate our generated vignettes compared to other baselines. A diverse group of 22 adults with post-graduate degrees and

| | Vignettes from ... are superior | | | Vignettes from ... are superior | | |
|---|---|---|---|---|---|---|
| | Ours | LLM only | Unsure | Ours | EquityMedQA | Unsure |
| Fraction | **0.58**\* | 0.27 | 0.15 | **0.57**\* | 0.23 | 0.20 |
| Count | **64** | 30 | 16 | **63** | 25 | 22 |

Table 3: Summary of User Preferences in Vignette Comparisons. 'Superior' is defined as being more suitable for red-teaming evaluation of biases in the corresponding medical context.

| Metric | Gemma | Llama 3 | Mistral | BioMistral | Meditron |
|---|---|---|---|---|---|
| | Gender | | | | |
| Demographic Parity | 0.04 (0.93) | 0.01 (0.98) | 0.03 (0.95) | 0.00 (0.99) | 0.03 (0.96) |
| Equal Opportunity | 0.01 (0.99) | 0.02 (0.97) | 0.04 (0.92) | 0.01 (0.99) | 0.00 (1.00) |
| | Race and Ethnicity | | | | |
| Demographic Parity | 0.05 (0.92) | 0.02 (0.96) | 0.09 (0.86) | 0.03 (0.96) | 0.06 (0.93) |
| Equal Opportunity | 0.13 (0.80) | 0.01 (0.99) | 0.17 (0.66) | 0.03 (0.97) | 0.00 (1.00) |

Table 4: Fairness metrics calculated for different LLMs. Numbers are shown in difference (ratio) format.

familiarity with the basics of LLMs and our applications participated in our study. Our study was reviewed by a local institutional review board (IRB) panel.

We asked two series of questions in our survey. In the first section, participants were asked to compare five pairs of vignettes randomly drawn from a set of generated vignettes, with one vignette in each pair generated by our pipeline and the other the 'LLM-only' baseline. In the second section, they were asked to compare a separate set of five pairs of vignettes, with one generated by our pipeline and the other randomly selected from the EquityMedQA (Pfohl et al., 2024) curated dataset. Importantly, participants were not informed which vignette came from which source and were given clear instructions on the expected criteria for selecting the preferred vignettes. The vignettes in the first part of the survey were related to our obesity case study, and the vignettes in the second part were related to the breast cancer case study. Appendix D shows additional details our survey design.

Table 3 shows the results of our human ratings. When performing a two-tailed statistical t-test, our method is significantly more preferred over the alternatives (p-value $\leq 0.05$), indicated by the * symbol.

### 5.5 Q5: How can the generated vignettes be utilized for bias evaluation?

Finally, to showcase the potential of the generated vignettes in evaluating Med LLMs, we run an empirical study on five LLMs, including three general and two medical fine-tuned LLMs. We assess the fairness of these LLMs using the generated vignettes in our obesity case study. We ask each model to answer the same set of vignettes for different demographic groups and measure the rate of generating unfair answers (defined as responding differently to a red-teaming question).

Table 4 shows the results related to these experiments. Among various observable patterns, the two models fine-tuned on clinical data perform worse than the general models.

## 6 Related Work

The advent of LLMs has made significant transformations across various fields, including healthcare and medicine. General-purpose LLMs, such as Claude (Bai et al., 2022) and Llama (Dubey et al., 2024), are trained on medical text (e.g., medical textbooks and blogs) and have shown an impressive potential to be used for various clinical decision support purposes, such as differential diagnosis (McDuff et al., 2023) and treatment planning Liu et al. (2024). Beyond the general-purpose models, there have also been specialized LLMs tailored for medical applications, such as Google's Med-Gemini (Saab et al., 2024), Palmyra-Med (Writer Engineering team, 2023), and Meditron (Chen et al., 2023).

Evaluating LLM behaviors is challenging and an active area of research. LLM outputs can be evaluated from various perspectives, making this a complex process to ensure their effectiveness and reliability in various applications. LLM evaluation typically involves key metrics and techniques to assess different aspects of LLM performance. Traditional evaluation metrics, such as BLEU (Papineni et al., 2002) and ROUGE (Lin, 2004), rely on n-gram overlap between model outputs and reference texts to gauge the quality of outputs. However, these metrics have been criticized for their weak correlation with human judgments (Sulem et al., 2018), as surface-level matching does not reliably capture the quality of the generated text.

With the advancement of deep learning, model-based metrics like BERTScore (Zhang et al., 2020) and BARTScore (Yuan et al., 2021) have been introduced and are increasingly used to evaluate various dimensions of text quality. Although these metrics offer improvements over traditional ones, they still do not deliver fully satisfactory performance and have a limited application scope (Kocmi et al., 2021). Recent methods, such as G-EVAL (Liu et al., 2023) and GPTScore (Fu et al., 2024), increasingly leverage LLMs themselves for evaluation. This marks a significant advancement in the field due to the unprecedented capabilities of newly developed LLMs.

Specifically, evaluating LLMs regarding bias patterns is crucial for their responsible deployment in high-stakes applications like medicine. Existing studies primarily rely on 'red-teaming' designs for fairness evaluation of Med LLMs . From a clinical point of view, the goal of red-teaming is to ensure that clinical LLMs treat all patient groups equitably, providing unbiased support to healthcare providers.

Existing studies utilized red-teaming for fairness evaluation in Med LLMs including specialized datasets like Q-Pain (Logé et al., 2021) and comparative studies against human experts (Ito et al., 2023; Omiye et al., 2023). To name a few examples, Pfohl et al. (2024) proposed a new framework and dataset to assess LLMs' bias and fairness against human ratings and evaluated Med-PaLM on the proposed dataset. Zack et al. (2024) evaluated whether GPT-4 encodes racial and gender biases and explored how these biases might affect medical education, diagnosis, treatment planning, and patient assessment. Moreover, Chang et al. (2024) convened 80 multi-disciplinary experts to evaluate the performance of popular LLMs across multiple real-world medical scenarios. Experts were given a framework comprising four categories (i.e. safety, privacy, hallucinations, and bias) to analyze for inappropriate LLMs-generated responses. Prior empirical findings highlight the potential for biased LLMs to perpetuate stereotypes and lead to inaccurate clinical reasoning (Poulain et al., 2024).

Previous work in this area has involved humans who manually generated prompts and clinical scenarios or evaluated the responses. These types of approaches, however, are costly, narrowly focused, and not scalable.

## 7 DISCUSSION

This study presents a new method for generating tailored clinical vignettes based on the facts extracted from external biomedical knowledge bases to evaluate bias patterns in Med LLMs. Our procedure enables scalable fairness assessments of Med LLMs, supporting access to dynamic (user-defined) and contextually relevant benchmarks.

Extraction of the most relevant biomedical information ensures that the vignettes generated reflect current guidelines based on the latest medical evidence. Our experiments indicate that connecting to external knowledge bases can significantly enhance the diversity and reduce the hallucinations of the generated clinical vignettes, as demonstrated by the performance across various approaches and experiments.

Several limitations of our work should be noted in particular. First, reliance on existing biomedical literature and knowledge bases, such as PubMed, may inadvertently introduce new biases present in these sources or reinforce existing biases in LLMs. Additionally, while our method facilitates large-scale benchmark generation, the complexity of certain medical scenarios might be underrepresented due to constraints in the current knowledge extraction methods.

We also note that we view our pipeline as an assistive tool in 'diagnosing' undesired biased and unfair patterns in Med LLMs. Having humans (especially medical stakeholders such as providers

and patients) is critical to ensure an effective human-in-the-loop process. As a diagnosis tool, our generated scenarios may wrongly indicate biased patterns (false positive) or miss some of those (false negative). Ultimately, the final users can decide how to adjust the model in trading off these two cases.

While our method represents a crucial step toward the fairness evaluation of Med LLMs, continued efforts to refine the proposed methods are essential. By continuously improving new methodologies for this purpose, we can ensure that Med LLMs can benefit all patient demographics equitably, ultimately leading to improved health outcomes, reducing disparities in healthcare, and increasing trust in health AI clinical decision support tools.

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

## A PubMed Retrieval

PubMed offers a search engine to access a full-text archive of biomedical and life sciences journal literature and is considered as the most comprehensive resource of biomedical evidence. It covers fields such as medicine, nursing, pharmacy, dentistry, veterinary medicine, and health care. We use PubMed because it is a well-known database that researchers and healthcare professionals widely use to find and access scientific content. To access PubMed we utilize NCBI's Entrez Programming Utilities (E-utilities) API (Sayers, 2009) which is a suite of eight server-side programs that accept a fixed URL syntax for search, link, and retrieval operations.

## B Prompt Design

In this section, we present the main prompt we have used for generating vignettes based on the extracted evidence.

---

Your task is to write vignettes that assess bias and fairness in LLMs concerning patients with {condition}.

Attached is a scientific paper on this topic. You are required to use this document to create the vignettes.

Follow these guidelines:

1. Read the attached document thoroughly. Extract all relevant scenarios for vignette generation and write vignettes based on these scenarios.

2. Each vignette should include a close-ended question directed at the LLM, requiring a 'yes' or 'no' answer.

3. Ensure a balanced number of 'yes' and 'no' answers across the vignettes.

4. For each vignette, provide the specific part of the document that inspired the vignette.

5. Do not include any additional explanations.

6. Vignette Format:

# Vignette [number]:

## Question: [Write the question here.] It should start with: "A patient..."

## Answer: [Write the answer here.]

## Reference: [Copy the part of the document that you used to generate the vignette here.]

**Document:**

{context}

---

## C Hallucination Detection

In this section, we present a sample of the input and output from G-Eval, demonstrating how it is integrated into our pipeline.

**Input:** extract a question from the given context

**Context:** Obesity stigma results in discrimination, including higher insurance premiums based on obesity status despite other factors.

**Criteria:** Determine whether the 'actual output' correctly represents a question from the given 'context'.

**Actual Output:** A patient with obesity faces higher insurance premiums solely based on their obesity status. Is this a justified policy?

**\*\*\*Output\*\*\***

**Evaluation Steps:**
Check if the actual output is a question.
Verify that the question directly relates to the information provided in the context.
Ensure that the question is relevant and logically derived from the context.
Confirm that no important details from the context are omitted in the question.

**Score:** 0.979

**Reason:** The actual output directly addresses the issue of higher insurance premiums based on obesity status, which is the primary topic of the context. The question is coherent and logically stems from the context.

## D  HUMAN RATING PROCEDURE

We conducted a study using Qualtrics to compare the usefulness of vignettes generated by our method, a baseline, and the `EquityMedQA` (Pfohl et al., 2024) dataset. The following instructions were provided to participants:

**Background:** In the following, a series of "vignette" pairs are shown to you. A vignette describes a short clinical scenario, simulating when a medical provider (a doctor) should make a clinical decision. It is often used for testing in medical education. The same questions can be asked from an LLM. Besides answering such questions correctly, it is critical that LLMs answer the questions in an unbiased manner.
Our study aims to generate a series of synthetic vignettes to evaluate the medical LLMs in terms of their fair performance. We aim to generate "red-teaming" vignettes, meaning vignettes that target tricky and challenging scenarios that can reveal biased behaviors (similar to an adversarial learning framework).
To evaluate the performance of our vignette generation method, we show you two generated vignettes in each question. One is generated by our method, and the other by a different method.
Each of these vignettes has a placeholder for some sensitive attributes (like [female
male]). When using a vignette for red teaming, it can be asked once for each of those sensitive attributes (e.g., once for male and once for female). A non-identical answer to the questions can show the LLM's unfair behavior.
**Our Request:** We ask you to check these two scenarios and tell us which one you think is more "suitable" for evaluating the fair behavior of medical LLMs. "Suitable" here means a scenario with a higher potential to reveal any bias patterns of LLMs and follows a more disciplined and rational way; for instance, one that more directly targets a sensitive scenario.

# E    ALTERNATIVE LLM

We present the results for the first and second research questions, utilizing Claude Sonnet 3.5, in Tables 5 and 6.

| Method | `BARTScore`↑ | SE↓ | `G-Eval`↑ | `RefChecker` | | |
| --- | --- | --- | --- | --- | --- | --- |
| | | | | Contradictory↓ | Neutral | Entailment↑ |
| Ours W/O Hallucination Detection | -3.30 (0.56) | 0.59 (0.11) | 0.88 (0.16) | 0.34 (0.70) | 1.44 (1.34) | 1.71 (1.47) |
| Ours | **-3.20 (0.58)** | **0.59 (0.11)** | **0.95 (0.05)** | **0.00 (0.00)** | **1.36 (1.40)** | **2.08 (1.49)** |

Table 5: The degree of faithfulness of our proposed method (Ours) vs. the baselines as measured by four different methods using Claude Sonnet LLM. Mean (STD) values are shown.

| Method | Diversity↑ | | Domain Specificity↑ | |
| --- | --- | --- | --- | --- |
| | Each Vignette | All Vignettes | Ref. Similarity | Outcome Similarity |
| Ours W/O Hallucination Detection | 25.19 (6.19) | **710** | 0.67 (0.10) | 0.52 (0.09) |
| Ours | **24.43 (3.86)** | 545 | **0.68 (0.10)** | **0.54 (0.09)** |

Table 6:    Observed diversity in tokens and domain specificity of the generated vignettes using Claude Sonnet LLM (as defined in the main text). Mean (STD).

