# OpenReview forum: "Enabling Scalable Evaluation of Bias Patterns in Medical LLMs"
_ICLR.cc/2025/Conference — Submitted to ICLR 2025_

### Official Review · Reviewer_6586 · 2024-10-22

**Soundness:** 2
**Presentation:** 2
**Contribution:** 2
**Rating:** 3
**Confidence:** 3

**Summary:**

This paper introduces a scalable evaluation protocol for determining biases in medical LLMs. The authors introduce a system for automatically generating clinical scenarios or "vignettes" that test for biased outcomes based on various sensitive attributes. This helps scale bias evaluations, which were previously manual and limited in scope. The system relies on external biomedical knowledge bases like PubMed to ensure that the generated vignettes are grounded in real medical evidence, minimizing hallucinations. The method explicitly ensures that certain medical scenarios that require specific treatments for sensitive attributes (like gender or race) are treated appropriately, without overgeneralizing or underrepresenting critical medical considerations.

**Strengths:**

- It appears their method is working much better than previous works.

**Weaknesses:**

- The writing can overall be improved. The ideas presented in this paper are not super new. Several preprints have come out demonstrating a similar type approach. (https://arxiv.org/abs/2409.16430, https://arxiv.org/abs/2404.15149, https://aclanthology.org/2024.bionlp-1.5/)
- structural organization of this paper is poor (related works should be introduced early to help provide the background of the research question being asked.)
- The method is weak. The ideas of retrieval, knowledge graphs integrated with LLMs are not particularly new. This is more of an application paper more than anything.
- In the methods they say they only use GPT4 then in section 4.5 they list a whole bunch of new LLMs for evaluation.
- Figure 2 caption is poor and not really describing what the figure is representing.
- the Vignette counts in general are tiny. I think it is only 100.
- paper does not have checklist

**Questions:**

1. The motivation behind the obesity and other tasks were not well motivated? A demonstration and wholistic evaluation on many standardized clinical scenarios can be helpful.
2. One of the massive limitations is that PubMed introduces more biases? Authors did not address how to overcome this.
3. Why did you only generate 100 vignettes?

---

> ### Author Response · Authors · 2024-11-28
>
> **Similar to prior work** We respectfully disagree with the reviewer on this note. The listed papers by the reviewer focus on different aspects. While these types of studies aim to run empirical analyses to evaluate bias patterns in LLMs, they actually further highlight the gap in the literature that our work aims to fill (i.e., running such evaluations at scale is not feasible now).  The listed papers (and similar ones) all use small benchmarking datasets. Our work aims to propose a new method to generate such benchmarks based on rigorous medical evidence. We are not aware of any prior work that presents a similar approach for scalable evaluation of medical LLMs.
>
> **Method is weak** Our proposed method for ensuring outcome independence using medical ontologies (UMLS) is novel and not present in any prior work. It is true that our proposed pipeline utilizes a variety of existing methods, but the way we combine them in our proposed method is novel. Several key elements are based on state-of-the-art methods in LLM research (such as the methods for minimizing hallucinations). We are not aware of any similar pipeline in the literature. Additionally, we believe the significance of our study stems not only from its methodological novelty but also from the urgency of this line of research in addressing the growing concerns about the responsible application of medical LLMs in practice.
>
> **Different LLMs in the Experiment** Please note that we use GPT-4 and Calude Sonnet for the generative LLM inside our pipeline (the LLM inside Figure 1). Any strong LLM can be used for this purpose. In Section 4.5, we use the generated vignettes to evaluate five target LLMs in medical tasks. Again, any LLM can be evaluated using the generated vignettes.
>
> **Small number of vignettes (100)** We used 250 vignettes for our human evaluations. We now generate three times more vignettes for the newly added experiments related to women’s health and cancer (besides the original obesity case study).
>
> ***PubMed can introduce new bias** This is a very reasonable concern that is raised. We aimed to minimize the biases in the very large and diverse set of PubMed articles by ​ limiting the retrieved articles to only clinical guidelines, systematic reviews, and clinical trial reports. Specifically, we ensured the most up-to-date guidelines published by the relevant professional associations (which often offer the golden standard for medical practice) are included.
>
> We have updated the paper to clarify the concerns raised by the reviewer additionally.

---

> > ### Comment · Reviewer_6586 · 2024-12-02
> > **Thanks**
> >
> > I thank the authors for their rebuttal. I however stand on my decision and decide not to change my score.

---

### Official Review · Reviewer_smbu · 2024-10-31

**Soundness:** 2
**Presentation:** 3
**Contribution:** 3
**Rating:** 5
**Confidence:** 4

**Summary:**

This work proposes a structured approach to generating clinical vignettes to use for red teaming medical large language models. The proposed approach is a pipeline that first extracts relevant articles from the clinical literature, which are then conditioned on to generate vignettes using a prompted LLM. The pipeline includes a hallucination detection procedure that ideally assesses the factuality of claims and faithfulness of the vignette to the source articles. A further step conducts an “outcome independence check”, which is motivated to filter vignettes to the set of outcomes and clinical contexts for which there is no association between the sensitive attribute and the outcome of interest such that the generated answers to pairs of prompts containing different sensitive attribute group identifiers can be compared. Experiments are performed to assess the performance of each of the components of the proposed approach.

**Strengths:**

* The limited scalability of data for red teaming and adversarial testing is a critical and unsolved problem. The core motivation for the work to generate vignettes for red teaming through grounding in the clinical literature and evidence for the presence of associations and/or disparities across patient groups is a strong one.
* The approach to grounding and hallucination detection is reasonable and well-executed.

**Weaknesses:**

* Clarity and validity of methods related to the outcome independence check
  * I found the description of the outcome independence check (section 3.4) difficult to follow, even with personal experience working with UMLS, such that it is difficult to assess whether the proposed approach is reasonable. I believe the ambiguity comes from the language, “we especially extract a subset (S_Anc) that belongs to the specified sensitive attributes (such as specific gender or ethnicity)”. Here, it is not clear what it means for a concept to “belong to” a sensitive attribute. Do UMLS terminologies contain annotations that a condition is associated with particular demographic groups? Alternatively, does UMLS contain annotations in cases where a demographic group is uniquely associated with a particular condition? This latter case does correspond more closely to the provided examples (e.g., prostate cancer, pregnancy, gestational diabetes), but it is not clear.
  * The work would benefit from a clearer description of the motivation and desiderata for the outcome independence check and for the desired properties of the generated counterfactual pairs. For example, it is not clear how the approach is intended to consider cases where there is some non-trivial population-level association between an outcome and a sensitive attribute (e.g., the condition occurs in all groups, but at different rates). It is also important to note that many of these differences are reflections of health disparities caused by differential exposure to social and structural determinants of health (e.g., differences in healthcare access and quality), with no causal biological basis. In some cases, but not all, allowing for different responses across groups may actually promote equity if it means that the negative consequences of structural inequity may be more readily counteracted. In other cases, it may be that there is some biological basis for disease associations (e.g., skin cancer by skin tone), but all groups There are, of course, deeper normative questions to address regarding each of these points. However, as is, the work leaves the motivation underspecified, and it thus unclear how to evaluate whether the approach addresses the intended goals and requirements.
* The analysis presented in section 4.5 is relatively superficial and difficult to interpret. Some specific issues that would improve the work if addressed:
  * It is not clear what the reference is that the outputs are being compared to. For example, for the comparison of outputs for racial identifiers, are the groups compared pairwise, or is there some comparison to an output that does not include identifiers?
  * The scope of the evaluation is relatively narrow, only testing one condition and a limited set of sensitive attributes. As the motivation for the study was to enable scalable evaluation, some effort to test whether that goal has been achieved would improve the paper.
  * It would be helpful to establish a baseline “self-disagreement” rate, assuming that the model outputs are non-deterministic. For example, if the disagreement rate is 0.15 for Male vignettes, what is the disagreement rate for Male vignettes following repeated calls to the model?
* Section 4.4: Human evaluation of vignettes
  * The human evaluation study is constructed in such a way that may bias the results towards the proposed approach over EquityMedQA (one of the main comparators). In my view, the main reason for this is because the current study generates questions conforming to a particular clinical vignette structure that does not conform to the intended structure for EquityMedQA questions. The instructions for the human raters (Appendix D) is then written in such a way to consider questions conforming to the structure of the generated as preferred. To be clear, EquityMedQA is a collection of several datasets, each generated through different methodologies, and designed to serve different purposes. Not all of the questions are intended to be well-formed clinical questions (e.g., some questions consumer or patient-oriented, or are intentionally malformed to facilitate adversarial testing). EquityMedQA questions also do not always explicitly contain a sensitive attribute, nor were they necessarily designed to support sensitive attribute augmentation and assessment of counterfactual fairness. These points are raised as properties of ideal vignettes in the rater instructions. Two of the datasets contained within EquityMedQA were actually designed specifically for counterfactual analyses like the one that the present work conducts, and those datasets may be the more relevant ones to compare to (the approach used to create those datasets may also be a relevant baseline for this work as a whole). As a further point, despite blinding, it may not be difficult for the raters to differentiate between the examples in the two datasets given the consistent structure of the proposed dataset (yes/no structure, topical clustering by obesity and and breast cancer).
* Minor issues
  * Line 305: “[...] the EquityMedQA dataset (Pfohl et al., 2024), which is a collection of seven datasets containing both human-authored by 80 medical experts and AI-generated medical queries [...]”. The part of this statement regarding 80 medical experts is not an accurate description of EquityMedQA. Consider rephrasing to simply “human-authored and AI-generated medical queries”.
  * Section 4.5, section title typo: “Bis” -> “Bias”
  * Framing of red-teaming: This work (arguably inaccurately) presents a particular form of counterfactual fairness assessments as the primary form of red teaming. It would be better to frame this form of evaluation as one of many approaches to red teaming, rather than the main one. See this review by Feffer et al for further reading (https://ojs.aaai.org/index.php/AIES/article/view/31647/33814).

**Questions:**

* The most critical issue in my assessment relates to the clarity, motivation for, and validity of the approaches used for constructing the outcome independence check and sets of counterfactual pairs, as discussed in the weaknesses section. Some specific suggestions:
  * Revise section 3.4 for clarity.
  * Consider adding a detailed example of how associations are identified using UMLS.
  * Clarify how counterfactual pairs are intended to be defined in cases with differences in disease incidence, addressing questions raised in the weaknesses section.
* Addressing concerns raised in the weaknesses section related to the empirical evaluation would improve my assessment of the work.
  * Clarify methodology for computing fairness violation rates in section 4.5.
  * Consider expanding the experiments to include a broader range of conditions and sensitive attributes to demonstrate scalability.
* Given the issues raised regarding the comparison to EquityMedQA, consider the following:
  * Refocus the comparison specifically on the counterfactual datasets and methodology for creating counterfactual pairs (e.g., using prompting) from Pfohl et al.
  * Consider expanding the diversity of conditions considered in the case study.

---

> ### Author Response · Authors · 2024-11-28
>
> We especially thank the reviewer for their detailed and in-depth review and suggestions. We have carefully followed the suggestions outlined in the items below. We have fixed several other minor issues listed by the reviewer.
>
> **Clarity, motivation, and validity improvement** We have revised section 3.4 as per suggestion. We have added a new example for UMLS usage (also below) and further improved the description of this process.
>
> For example, Pregnancy (C0032961) is referenced in 25 different vocabularies included in the UMLS. To analyze outcome dependence, we explored the neighbors of the "pregnancy" node within these vocabularies, extending up to three levels. On average, 15.3 nodes are identified per vocabulary containing "female," whereas no nodes contain the keyword "male." The pipeline uses this as a proxy to associate "pregnancy" predominantly with female-specific contexts within the analyzed vocabularies, reflecting the biological and societal understanding of pregnancy as a female-exclusive phenomenon.
>
> The reviewer highlighted several very accurate considerations about automating bias evaluation in Med LLMs. In fact, the highlighted notes were among the descriptions we ended up not including in the submitted version due to space limitations. More specifically, in developing our pipeline, we err on the side of caution by filtering out any base vignette that the independence between the target health outcomes and sensitive attributes is not fully established. This means that our pipeline will not cover the scenarios with various documented prevalences among the subgroups (due to any biological or social roots). This step occurs during the Hallucination Detection stage. This type of design is a limitation for our pipeline but allows the generated vignettes by the pipeline to be more reliable and consistent for evaluating counterfactual fairness.
>
> We also note that we consider our pipeline as a screening tool aiming to support a human-in-the-loop process. Our design aims to trade off false positives (LLM not truly biased in a scenario but flagged as such) over false negatives.
>
> We updated the text in a few places to clarify the focus area of our pipeline and its intended usage for medical scenarios with established independence between the health outcomes and sensitive attributes.
>
> **Evaluation improvement** We have updated Section 4.5, run entirely new experiments, and replaced the corresponding table (Table 3). We are now reporting more standard experiments of measuring group fairness across subgroups using the generated vignettes.
>
> Additionally, we have now expanded our experiments by including a series of two case studies related to women’s health (pregnancy-related vignettes) and cancer (breast cancer).
>
> **Concerns about the usage of EquityMedQA** We indeed only selected a subset of vignettes from the two datasets inside EquityMedQA that matched the same style as our approach (i.e., probing counterfactual fairness). We further highlighted this aspect, along with the similar prompting mechanism we use for creating counterfactual pairs. The “sensitive attribute augmentation” part of our pipeline (section 3.5) is aligned with and inspired by the approach used by Pfohl et al.’s study.
>
> Moreover, as noted above, we are now significantly expanding the diversity of our conditions and have included two series of new experiments in our study.

---

> > ### Comment · Reviewer_smbu · 2024-12-02
> > **Response to rebuttal**
> >
> > Thank you for the clarifications and updates to the work. Overall, the clarifications are helpful and improve the paper. I have adjusted my score to a weak reject (5). Here is why my score is not higher:
> >
> > * While the revisions and included example are helpful, the methods section describing the outcome independence check could still be improved. The text is still relatively unclear, with key details only explained in the pregnancy example and not in the methodological description preceding it. For example, it is not clear what it means for a concept “to belong to the specified sensitive attribute” (line 304), even if this is explained as cases where the node text contains the string for the sensitive attribute in the later example. Relatedly, it is unclear what the “full” and “partial” relationships between the sensitive attribute and outcome refer to. The equation could also use some work, as S_Input and S_Ancestors are not defined (the text defines S_Anc but not S_Ancestors). It is also not clear if this equation faithfully reproduces the written algorithm. For example, my interpretation was that cases where no terms related to the sensitive attribute were identified would be a “full relationship”, and thus a case where S_Red-Team should include S_Input, but the equation as written would instead return S_Ancestors, which is empty.
> > * A further potential concern regarding the validity of the outcome independence check is that it may be more appropriately framed as a procedure that rejects cases for which there is clear biological outcome dependence (e.g., pregnancy, breast cancer, and prostate cancer), but likely does not reject cases for which outcome dependence is weaker or due to social or structural causes (e.g., cardiovascular disease by race and ethnicity). However, the work argues that all cases that pass the outcome independence check “are safe to be augmented”, assigning confidence to the cases that are not rejected. In other words, if we think of the outcome independence check as a screening algorithm and the cases for which outcome independence is actually violated, the algorithm has high specificity (low false positive rate) and low sensitivity (high false negative rate, whereas it is argued that the procedure has high sensitivity.
> > * The new Table 4 is easier to read and clarifies my concerns with the metrics, although the way that demographic parity and equal opportunity are computed is not described in the text (as far as I can tell). I am familiar with these metrics in general, but it is not clear how they were computed here.
> > * Thank you for the clarifications regarding EquityMedQA.

---

### Official Review · Reviewer_sXia · 2024-11-03

**Soundness:** 3
**Presentation:** 3
**Contribution:** 2
**Rating:** 5
**Confidence:** 3

**Summary:**

This paper presents a pipeline for automatically generating clinical vignettes to evaluate bias and fairness in medical Large Language Models. The key innovation is a pipeline that combines biomedical knowledge retrieval from PubMed with controlled vignette generation and multiple validation steps to ensure quality. The method specifically addresses three main challenges: domain-specificity of fairness evaluation in medicine, dependencies between health outcomes and sensitive attributes, and hallucination in generation. The authors demonstrate the effectiveness of their approach through comprehensive experiments and provide a new dataset for fairness evaluation in the domain of obesity treatment.

**Strengths:**

The paper presents a new tool for automatically generating clinical vignettes to evaluate bias in medical LLMs, addressing a key challenge in the field.
The proposed pipeline incorporates multiple components to ensure the generated vignettes are evidence-based, domain-specific, and reduce hallucinations.
The use of biomedical knowledge bases and ontologies helps ground the vignettes in established medical evidence and relationships.
The method includes checks for outcome independence and hallucination detection, improving the quality and reliability of the generated scenarios.
The experimental results demonstrate improvements over baseline methods in terms of faithfulness to medical evidence and diversity of generated content.
The work contributes a new dataset for evaluating fairness in medical LLMs, which could be valuable for future research in this area.
The proposed method enables more scalable evaluation of bias in medical LLMs compared to manual vignette creation, while maintaining quality.

**Weaknesses:**

Limited Scope of Case Studies: While obesity is a good starting point, the paper would be stronger with more diverse medical conditions to demonstrate generalizability. The current focus on a single primary case study leaves questions about how well the method extends to other medical domains.

Validation of Medical Accuracy: While the paper uses various computational metrics for evaluation, there's limited validation of the medical accuracy of generated vignettes by practicing clinicians. The human evaluation focuses more on fairness evaluation utility than medical correctness.

Bias intrinsic: There is limited discussion on potential biases introduced by the information retrieval process from PubMed and the PubMed Knowledge Graph as we know there are healthcare disparities in real life and medical knowledge systems.

Retrieval: The method for identifying relevant articles seems simplistic and could benefit from more sophisticated relevance ranking algorithms. Further there are concerns that due to constant shifting medical knowledge that the system could retrieve out of date information in the vignettes.

Scalability Analysis: The computational costs and time requirements of the pipeline aren't thoroughly analyzed, particularly the knowledge retrieval and validation steps which could become bottlenecks at scale.

Minor:
- typo line 448-- Bis --> Bias

**Questions:**

1. Clinical Validation and Accuracy-- Could you provide a pilot evaluation of medical accuracy by having a panel of practicing physicians review a representative sample (e.g., 50-100) of generated vignettes across different medical conditions? Please include quantitative analysis of error rates and types of medical inaccuracies identified.

2. Domain Generalization and Scalability-- How well does your method generalize beyond obesity? Could you validate the pipeline on at least 1-2 additional medical conditions with known disparities (e.g., cardiovascular disease, psychiatric conditions) and provide performance metrics for each stage of the pipeline across these different domains?

3. Bias Propagation and Mitigation-- How do you ensure that biases present in PubMed and medical knowledge bases don't get propagated or amplified in the generated vignettes? What specific safeguards or validation steps in your pipeline address this risk? Have you evaluated this component of the retrieval?

4. System Performance and Computational Requirements-- What are the end-to-end computational requirements and processing times for generating validated vignettes? Please provide a detailed breakdown by pipeline stage (retrieval, generation, validation) and discuss how these requirements scale with different medical domains and knowledge base sizes.

5. Medical Knowledge Currency-- Given the rapid evolution of medical knowledge and guidelines, how does your system ensure the generated vignettes reflect current medical best practices? What mechanisms could be implemented to identify and update vignettes when medical knowledge changes automatically?

---

> ### Author Response · Authors · 2024-11-28
>
> **Include results from a medical panel** We have been trying to organize the suggested panel of physicians by reaching out to the clinical experts in our network. We have had an obesity expert (a practicing MD) so far who has evaluated a subset of our generated vignettes and confirmed the appropriateness of those for the obesity case study (two suggestions for a pool of ~50 vignettes). Due to the logistical complexities of such a new human study (including providers’ availability, cost, and updated IRB), we have not yet formed a larger panel of physicians. However, we anticipate we will be able to report the results for a group of (at least four) physicians within one month (before January 2025).
>
> **Generalization beyond obesity** We have now included a new series of experiments related to two additional case studies, including women’s health (pregnancy) and cancer (breast cancer).
>
> **Bias propagation** While PubMed contains a vast collection of many diverse articles, we aimed to minimize the biases sourced from PubMed retrieval by limiting the retrieved articles to only clinical guidelines, systematic reviews, and clinical trial reports. Specifically, we ensured the most up-to-date guidelines published by the relevant professional associations (which often offer the golden standard for medical practice) are included. We have also updated the article’s text to highlight this more clearly.
>
> **System performance and compute requirement** We are now including a high-level analysis of the computational requirements for our pipeline. Notably, for each of the generated vignettes, our pipeline requires around 150,000 tokens. This is when using a pre-trained knowledge graph and vector space and based on an average length of 5,000 words per retrieved article when around 10 articles are retrieved. For a given medical domain, generating a reasonable-size dataset (~10K vignettes) should scale in a manageable way.
>
> **Ensuring up-to-date medical guidelines** Given the automated nature of our pipeline, end-users would be able to update the generated vignettes over time. In particular, the knowledge graph component of our pipeline supports dynamic updates, allowing it to stay current with respect to the latest developments in the field. Additionally, we prompt the generator LLM to favor retrieving the most recent articles. We thank the reviewer for bringing up this important and (what we believe) a strong aspect of our pipeline. We also added a note about this to the Discussion section.

---

### Official Review · Reviewer_jMD5 · 2024-11-04

**Soundness:** 2
**Presentation:** 2
**Contribution:** 2
**Rating:** 5
**Confidence:** 2

**Summary:**

The paper presents a method to evaluate bias in medical LLMs by generating clinical vignettes that force LLMs to respond across demographic groups. (Ideally having outcome independence of different types of demographics.)
Automated vignette generation is performed through information retrieval of biomedical knowledge bases like UMLS and PubMed, and filtered for hallucinations via RefChecker and G-Eval.

**Strengths:**

- The method allows for large-scale evaluation, generating a variety of vignettes quickly without human effort
- Retrieval of relevant literature and entities should help reduce hallucinations
- Additional hallucination reduction using recent techniques (RefChecker and G-Eval) is encouraging

**Weaknesses:**

- The method may still struggle with the nuanced complexities and biases of clinical cases, e.g. with certain races having a higher prevalence of certain symptoms (E.g. asian flush)
- The method could inadvertently reinforce existing biases present in biomedical literature. Since it relies on pre-existing knowledge bases, any systemic bias in these resources may propagate through the generated vignettes, leading to an inherent limitation in the bias evaluation framework.
- The human evaluation seems weak, with only 110 total counts of the User Preferences in Vignette Comparisons being evaluated. Additionally, it's interesting that LLM only seemingly outperforms / performs around the same as the ground truth from EquityMedQA. Could the authors discuss why this may be the case?

**Questions:**

See weaknesses.

---

> ### Author Response · Authors · 2024-11-28
>
> **Struggle w/ nuanced complexities** The complexities that the reviewer brings up are indeed relevant and important. Our pipeline, however, filters out any of such scenarios and only keeps the medical scenarios (vignettes) with established independence between the sensitive attributes and health outcomes. In the Discussion section, we explicitly acknowledge the intended usage of our pipeline (or its limited scope).
>
> Furthermore, we consider our pipeline as a “screening tool,” supporting scalable and automated evaluation of bias patterns in Med LLMs. Accordingly, our design does not claim it can guarantee a Med LLM is fair, rather it aims to identify concerning (biased) patterns for further human evaluation.
>
> Having said that, we still believe a pipeline connected to external medical evidence (i.e., through PubMed or ClinicalTrials.gov) would be able to extract many other nuances that have been noted in the community and use them for vignette generation.
>
> We have expanded our Discussion to capture the reviewer’s note.
>
>
> **Possible reinforcing of existing biases** We aimed to minimize the biases in the very large and diverse set of PubMed articles by ​ limiting the retrieved articles to only clinical guidelines (such as annual practice guidelines by the American Heart Association), systematic reviews, and clinical trial reports. Specifically, we ensured the most up-to-date guidelines published by the relevant professional associations (which often offer the golden standard for medical practice) are included. While such guidelines and recommendations may change over time, they capture the latest evidence-based knowledge about each specific medical domain.
>
> As noted in the previous comment, our pipeline does not intend to guarantee a Med LLM is 100% unbiased or fair and does not aim to mitigate identified bias patterns. Accordingly, even if the pipeline fails in some scenarios, it should not “reinforce” existing biases.
>
> We added a note to the discussion to clarify the concern raised by the reviewer.
>
>
> **Human evaluation clarification** Besides larger scale evaluation of our pipeline using LLM-based methods, we have included ~250 vignettes in our human evaluation. We have also expanded the number of studied case studies from one (obesity) to three (women’s health and cancer).
>
> (We assume the reviewer was referring to comparing our pipeline against EquityMedQA, as we did not compare the LLM-only method against that). Prior to human evaluation, our hypothesis was that the generated vignettes by our pipeline would have relatively similar (not worse) performance compared to EquityMedQA. The results (i.e., ours being preferred over EquityMedQA) were surprising to our team as well. As suggested by the reviewer, we have expanded the Discussion section to further clarify this. Overall, it is possible that our human raters were biased toward the generated vignettes (possibly due to using controlled formats for generating those), but the fact that our vignettes are comparable to human-generated vignettes is the key take-home message from the experiment.

---

> > ### Comment · Reviewer_jMD5 · 2024-12-02
> >
> > Thank you for the responses--I am still quite concerned with the low number of evaluated vignettes, and as such will keep my score.

---

### Author Response · Authors · 2024-11-28

We thank the reviewers for their time and for sharing their suggestions and constructive feedback. We provide point-by-point responses to the individual concerns below.

---

### Meta-Review · Area_Chair_6MnR · 2024-12-21

**Metareview:**

The paper presents a framework for generating clinical vignettes, motivated by scalable bias evaluations. In several medical settings, the paper examine the quality of these generated vignettes including across faithfulness, domain specificity, and bias under counterfactual fairness

Strengths (based on reviewers' input):
 - Live demo and reproducibility code are provided
 - Inclusion of relevant literature listed as an asset

Weaknesses
 - Vignette generation pipeline does not capture complexities of clinical cases. Additionally, the connection between the medical accuracy of the vignettes is not properly examined.
 - Risk of reinforcing biases in biomedical literature
 - Insufficient human evaluation (low number of samples and no medical panel of evaluators)
 - Lots of reviewer concerns about clarity, including confusing descriptions of the methodology and the experiments

Given the relatively simplistic setup and shallow evaluation protocol, I am recommending reject for this piece. Because the work is promising despite these concerns, I encourage the authors to address the reviewer concerns, and I look forward to seeing this work in a future research venue.

**Additional Comments On Reviewer Discussion:**

Reviewers raised points about clarity of the work, small number of evaluation samples as well as the lack of medical panel for evaluation, and the pipeline setup. Authors provided more details about the evaluation (increasing to 250 points) and explaining the barriers to assembling a medical panel for evaluation. Similar to the reviewers, I was not swayed by the rebuttal and find the concerns listed quite serious.

---

### Decision · Program_Chairs · 2025-01-22

Reject